# Impact of Long-Term Floods on Spatial Dynamics of *Myrmica scabrinodis*, a Host Ant of a Highly Threatened Scarce Large Blue (*Phengaris teleius*)

**DOI:** 10.3390/insects14110891

**Published:** 2023-11-18

**Authors:** Mitja Močilar, Klemen Jerina, Rudi Verovnik

**Affiliations:** 1Department of Biology, Biotechnical Faculty, University of Ljubljana, Jamnikarjeva 101, 1000 Ljubljana, Slovenia; rudi.verovnik@bf.uni-lj.si; 2Department of Forestry and Renewable Forest Resources, Biotechnical Faculty, University of Ljubljana, Večna pot 83, 1000 Ljubljana, Slovenia; klemen.jerina@bf.uni-lj.si

**Keywords:** species decline, myrmecophily, Slovenia, recolonization, spatial dynamics

## Abstract

**Simple Summary:**

Myrmecophilous butterflies are highly specialized insects that depend on specific ant species in their habitat to complete their life cycle. The present study was triggered by the drastic decrease in the population size of a myrmecophilous butterfly Scarce large blue (*Phengaris teleius*) between the years 2008 and 2012 in Ljubljansko barje in Slovenia. Our initial hypothesis was that the extensive and long-lasting flood in the fall of 2010 eradicated host ant species and consequently also decimated the *P. teleius* population. To test our hypothesis, we conducted spatial analyses that included the current distribution of *Myrmica scabrinodis* ant species, the extent and duration of three major floods since 2008, the distance from the nearest refuge for ants, and the areas where *P. teleius* was present in 2008 as an indicator of the presence of the host ant species at that time. We showed that prolonged flooding eradicated ant colonies in flooded meadows. In addition, we calculated the rate of recolonization to be approximately 29 m per year, which compensates for 1.8 days of inundation.

**Abstract:**

Extensively used wet meadows with high species diversity are under threat in Europe by anthropogenic pressure. The increasing frequency of prolonged flooding is emerging as an additional threat to this fragile environment. In our study, we investigated how prolonged flooding affects the spatial distribution and temporal dynamics (through mortality and recolonization process) of the host ant species *Myrmica scabrinodis*, which is essential for the survival of the endangered Scarce large blue (*Phengaris teleius*). The study was conducted in the flood-prone Ljubljansko barje plain situated on the southern edge of the species’ global range. Prolonged flooding in the study area, possibly affecting the past and current distribution of the host ant *M. scabrinodis*, was recorded in 2010, 2013, and 2017. In 2020, we set 160 ant traps to estimate the distribution of host ants in a system of meadows covering the entire gradient of flood history. Results indicate that *M. scabrinodis* survives the flooding for up to three days, starting to disappear if flooding persists longer. After the flooding recedes, ants gradually recolonize empty habitats from the surrounding upland refugia. Our spatial analyses predict that the average recolonization speed was about 29 m per year and that in a year, ants compensate for the mortality effects of 1.8 days of flooding by recolonization in a year. These results show that flooding should be considered as an additional (in some areas, a major) threat to the endangered *P. teleius* through its deleterious effects on the host ant species.

## 1. Introduction

Floods, like other natural phenomena, have a strong influence on the evolution of organisms. Floods play an important role in natural selection and the evolution of morphological and behavioral adaptations of plants and animals living in floodplains. Roughly speaking, natural disasters can be divided into smaller, more frequent events that alter the behavior of organisms and trigger multigenerational adaptation to living conditions through natural selection, and larger, less frequent natural disasters, causing the extinction of one or more species, thereby accelerating the evolution of surviving species [1]. Flooded habitats are considered to be ecosystems with one of the highest species diversities on Earth, which is due to the high evolutionary pressure caused by regular disturbances in this demanding environment [2]. The fauna and flora of these ecosystems have evolved as a result of hydrological gradients, floods, and different types of soil, resulting in a great diversity of plants, fungi, and animals. However, humans have indirectly and directly altered the flood regimes across Europe dramatically over the past 60 years, rendering these habitats among the most threatened in Europe [3].

The construction of dams, embankments, draining of floodplains, deepening of riverbeds, fortification of banks, and building of infrastructure, have a great impact on floodplains, which often cease to flood or, on the contrary, remain flooded for prolonged periods of time [4]. Moreover, climate change causes longer and more irregular floods, to which local vegetation and fauna cannot adapt [5]. The changing regimes and duration of floods directly affect the composition of ecosystems [6], which reduces the plant and indirectly the animal diversity of such areas. Terrestrial animals are most at risk of drowning during floods when they are submersed in poorly oxygenated standing water. They are also threatened by the possibility of being flushed from their habitat, and by the impact of toxic substances that form in the soil from decaying vegetation such as methane, acetic acid, and butyric acid. With the presence of sulfates (fertilizers, pesticides), the formation of sulfuric acid can also occur [7]. All these threats only increase with the prolonged duration of floods and ultimately cause local extinctions of maladapted species. 

Ljubljansko barje Plain in Slovenia has a long history of anthropogenic interventions, particularly drainage, aiming at the prevention of floods in populated areas. In addition, due to climate change, extensive precipitation has become more frequent in the region [8], causing longer flooding of low-lying areas along main rivers, especially in the autumn (Figure 1). Far-reaching potential effects of prolonged flooding in Ljubljansko barje were first noted in 2012 when a sharp decline in abundance and distribution of the Scarce large blue (*Phengaris teleius*) was documented during regular monitoring of the Habitats Directive species [9,10,11,12]. The population and suitable habitat of *P. teleius* have not recovered to its former scale ever since these floods [12]. The decline coincides with the 2010 long-lasting autumn flood, which persisted for up to 29 days in the surveyed area [8]. No specific research was conducted at that time to identify the reasons for the sharp decline but direct mortality of larvae and pupae due to long-lasting flooding and the retreat of the host ants from the flooded areas were speculated as potential factors. 

The main hosts of the *P. teleius* in Central Europe are *Myrmica scabrinodis* and *M. rubra* ants [13], but as Tartally et al. [14] show *Phengaris* species are highly adaptive and can switch or share other *Myrmica* hosts according to their availability. However, in the study area, *M. scabrinodis* was identified as the sole host ant species [12]. The caterpillars of the first three instars feed with inflorescences of the Great Burnet (*Sanguisorba officinalis*), which is their sole host plant [15,16]. The development of caterpillars on the flowers usually takes about 3 to 4 weeks [15]. After the transition to the fourth instar, they descend from the host plant and begin to emit acoustic and chemical signals [17] which attract and confuse host ants, so that they adopt them as their own brood [15,16]. The caterpillars are predatory on ant brood and their development takes from 10 to 22 months depending on the size of the ant colony [18]. After pupation in early summer, adults emerge from the ant nests at the end of June or the beginning of July depending on the season [19].

Although the effects of flooding on ants are not well studied, several adaptations that allow ants to survive floods have been described [20]. The *Myrmica* ants, in particular, are particularly exposed to flooding as they build underground nests. Representatives of this very widespread genus have developed two strategies that allow them to survive several days of inundation. The first strategy is to use the trapped air bubbles in the anthill during the flooding; they also tolerate increased concentrations of carbon dioxide. The species is able to take advantage of the air bubbles that form in the anthill gallery and with their help spend several days completely underwater. If the floods do not recede for a longer time, the concentration of carbon dioxide in the bubble rises to the limit value causing ant mortality [21]. Another survival strategy is swimming and finding a refuge on dry land. Representatives of the genus *Myrmica* have often been observed swimming several meters to land or the nearest branch [22,23]. Even less is known about *Myrmica* ants’ recolonization of suitable habitats, but it likely appears through budding from the existing colonies [24] as the nuptial flights, especially of females, are very localized [25].

Our aim was to study the effects of flooding on host ant *M. scabrinodis* distribution and recolonization. Specifically, we wanted to investigate (i) how floods and their duration affect the presence—the probability of survival of the *M. scabrinodis* host ant. We assumed that using the above-described strategies ants survive shorter floods, but with the prolongation of flood duration, the probability of survival decreases and finally drops to zero resulting in local extinctions. (ii) Whether and how the recolonization of host ants to unoccupied areas is proceeding after the end of floods. Habitat recolonization in mobile species occurs in at least two manners, either by gradual peripheral expansion of the population range, as is the budding of ant colonies, or by long-distance dispersal and the establishment of new (also remote) metapopulation nuclei, as is the case in nuptial flights in social insects. Since ants are less migratory for most of their lives (except for the nuptial flights, which are not common in *Myrmica* species), we expect that the first of both listed systems of recolonization would prevail. Therefore, we predict (i) that the probability of occurrence of ants decreases with distance from the nearest topographic refuge (elevated site enabling ant survival during floods), and (ii) probability of occurrence simultaneously increases over time (elapsed since the last major flood) due to the (gradual) process of recolonization. These patterns of mortality and recolonization are of great conservation importance also for the *P. teleius*, as it fully depends on the presence of its host ant.

## 2. Materials and Methods

The study area is located in central Slovenia south of the capital city Ljubljana in the large floodplain of the Ljubljanica River close to Bevke and Blatna Brezovica villages (Figure 1) [26]. The predominant land use is agriculture with pastures and hay meadows being the most common type. We used a detailed agricultural land use vector map provided by the Ministry of Agriculture, Forestry and Food to extract information on the current and past use of the grassland habitats in the area [27] and selected those with land use type “extensive grasslands” or “pastures” (i.e., potentially suitable for ants). Despite an extensive drainage network, these grasslands retained the humid floristic character, and the larval host plant of *Phengaris teleius*, the Great Burnet (*Sanguisorba officinalis*), is still widespread in the area. The study area is included in long-term national monitoring of the *P. teleius* populations [9,10,11,12,19,28]. Therefore, past distribution data for this species are available. The distribution of *P. teleius* before the first floods (Figure 1) could be equivalent to the minimum past distribution range of its host ant *M. scabrinodis* which was however not directly studied during the aforementioned surveys.

We used altitude raster layers (horizontal spatial resolution 12.5 m; vertical precision 1 cm; lidar derived) and daily water level data for Ljubljanica River at Vrhnika town measuring station (E 14.30196; N 45.97004) obtained from the Slovenian Environment Agency [29] to determine the past flooding regimes in the studied area. We calculated and visualized the extent and timing of the floods by subtracting the water level and the altitude of the measuring points from the altitude raster of the study area for each day with QGIS [30]. The procedure was repeated for all days with floods from the beginning of 2010 to the end of 2019. Only floods lasting longer than 10 days were taken into account for the planning of ant monitoring fieldwork, as such floods are expected to cause total mortality of *Myrmica* ants [22]. Three such longer flooding events were identified; the most extensive in September and October 2010, followed by intermediate in March and April 2013, and the least extensive in December 2017 (Figure 1).

The presence of *Myrmica* ants was surveyed on thirty-two separate meadow plots (size of plots from 0.025 ha to 4.92 ha; average 1.27 ha) selected to cover the entire flooding regime gradient (Figure 2). The minimum distance between two neighboring meadows was 0.5 m. Six meadows were never flooded for more than six days in the last decade and represented the control sites. Among flooded meadow plots, eight were last time underwater for a longer period in 2010, 10 in 2013, and eight in 2017 (Figure 2).

We set up 160 ant traps cumulatively on 32 separate meadows (5 traps per meadow) in September and October 2020 at the end of the mowing season. We used standard pitfall traps made of plastic pots dug in to level the rim with the ground and a metal cover roof to prevent direct rainfall. The fixative used in the traps was a non-attractant propylene glycol. The traps were marked individually and left for 72 hours at the site. We placed five traps at each meadow in a straight line (usually on the longest diagonal of a meadow) for their easier retrieval. The content of the traps was moved to tubes marked with the same code as the traps and examined for *Myrmica* ants using binocular loupe Leica M165C. The key from the AntWeb website [31] was used for species determination. All ants were identified at the species level.

To estimate the speed of recolonization of the area, we determined the boundaries of refuges for each flood, from which ants could inhabit the unoccupied areas after the floods withdrew. The following two criteria were taken into account when determining the polygons of refuges: (1) refuges have suitable habitat characteristics for the target genus of ants. To evaluate the habitat suitability of the sites, we used the already mentioned detailed map of the use of agricultural land [8]. (2) From a detailed map of altitudes. For each of the three floods, we extracted polygons of areas that were flooded for less than 10 days. This period was used as a criterion because ants are unlikely to survive longer floods [22]. We prepared a vector map of potential refuges for each of the three flood periods by intersecting the previously described maps.

For each trap, we checked the following set of variables with spatial queries and the above-described survey of ant presence: (1) the number of ants of the genus *Myrmica* in the trap (continuous value). (2) Duration of the flood in days, for each of the three floods separately (three continuous variables). (3) Distance from the edge of the nearest refuge to each ant trap (in meters, for each flood event separately); for the traps set outside refuges the distance is a positive value and for the traps within refuges the values are negative (three continuous variables). (4) Meadow plot ID. (5) Meadow plot usage in the flood year (three attributive variables: mowed meadow, cultivated area, pasture). (6) Time elapsed from each flood to the setting of the ant traps (in years, continuous variable). More detailed descriptions of variables can be found in the Appendix A.

### Statistical Analyses

We used histograms and rank correlation analysis to explore the principal characteristics of the dependent variable and all independent variables and their mutual correlations. Based on these findings we selected appropriate subsequent statistical analyses. The variable number of captured ants in traps had a significantly right-skewed asymmetric distribution. Therefore, all subsequent analyses used the binary variable presence of ants in a trap (yes/no; 0 ants = no, ≥1 ants = yes) as the independent variable.

In the first series of analyses, we examined how the impacts of each of the three floods dissipated over time due to ant spatial recolonization. For each separate flood, we fitted the logistic regression function to the ant binary occurrence data which describes the probability of ant presence with regards to flood duration (in days; Figure 3; left) or distance from the nearest refuge (in meters; Figure 3; right). To improve the comparability of results, all three fitted logistic functions depicting the probability of ant presence were visualized on the same graph, once for the variable flood duration (in days) and the second time for the variable distance to the nearest refuge (Figure 3). This analysis implicitly assumes that the effects of flooding at each location depend only on each individual flood, although this is not guaranteed as some locations have been flooded several (up to three) times.

In the second series of analyses, we estimated the speed of spatial recolonization of habitat and the time needed for ants to “compensate” for the effects of one day of flooding. To this end, the logistic regression function which predicts the probability of ant presence on the trap site with regard to distance to the refuge (or days flooded) and time from flooding to sampling (in years; continuous variable) was fitted to ant binary occurrence data. Recolonization speed was afterward calculated by dividing both estimates of function coefficients (parameter estimate) of the logistic regression model. The analysis was first done for the variable distance to the refuge and then for days flooded; in both cases, the second independent variable was time from the flood. Because the main purpose of these two analyses was to estimate recolonization speed, we did not require the independent variables in the logistic regression model to be significantly related to the dependent variable. This analysis (unlike the previous one) assumes that the effects of floods (up to three) on the survival of ants at a given location are cumulative, which seems intuitively very likely. Although this method does not guarantee that the effects of floods are quantified in a perfectly independent way, we are not aware of any better method. Unfortunately, the conditions in the study area also do not allow for a better set-up of plots to ensure that the effects of individual floods on the observed survival of ants are completely independent.

In the third set of analyses, we analysed which of all possible explanatory models (combination of explanatory variables) best predicts the probability of ant presence in traps. To do this, we used generalized linear mixed models (GLMM): the binary variable of ant presence in traps (logit link, binomial error distribution) was used as the dependent variable, and the duration of each of the three floods, distance from refuge for each of the three floods, habitat use, and time from each of three floods (in years; continuous) were used as independent variables. The main effects of independent variables and selected meaningful interactions (e.g., time from flood × days flooded) were included in the models. The variable meadow ID was treated as random and the other variables as fixed factors. The model best supported by the data was selected using the Akaike information criterion (AICc) and best subset algorithm, whereby potential models were limited in size to four included variables (main effects or interactions). A more detailed comparison of competing GLMM models can be found in the Appendix A.

All statistical analyses were done using Statistica 8.0 (StatSoft, Inc., Tulsa, Oklahoma) and R (version 4.1.3), package lme4 [32]. A *p*-value of <0.05 was considered statistically significant. 

## 3. Results

Of the three floods, the strongest one (lasting the longest period) was in 2010, followed by the 2013 flood; the 2017 flood was the mildest. The basic statistics of all three floods are given in Table 1.

A total of 292 *M. scabrinodis* ants were captured on 160 trap sites, or 1.82 per trap (CI 95% = 1.2–2.4). In more than 60% of traps (98/160) not a single ant was captured, followed by traps with one captured *M. scabrinodis* ant (9%); the highest number of ants in a single trap was 34. Of the 32 separate meadow plots, there was not a single capture on 14 (44%) plots, while captures in all traps were recorded on six (19%) plots. Twelve meadow plots (37%) had *M. scabrinodis* in at least one trap.

Independent variables (duration of flood and distance to refuge for that flood) correlate, which was expected. They also correlate with the dependent variable. The strongest correlation is with the number of ants and distance to the nearest flood refuge in 2013 (r = −0.58); the correlation with the duration of flooding (in days) in the same year is only slightly weaker (r = −0.57,) and both these variables are strongly correlated (Table 2).

Results of logistic regressions (first set of analyses) show that the number of days flooded and the distance of the trap site to the refuge edge both impacted the probability of ant presence on sites of traps in all three floods (Figure 3). However, the strength of correlations with dependent variables between floods are not the same. The longer the time after the last flood, the superficially longer duration of flooding the ants are supposed to tolerate and, conversely, they are more sensitive the more recent the flooding. This is related to ants having more time for recolonization after more distant (further in the past) floods, so these “differences” between flood events are an artifact of a longer longer-lasting colonization process due to the longer time available. According to the prediction for the 2017 flood (last flood), ants should be able to tolerate roughly 10 days of flooding, whereupon they should disappear; the results of analysis for the older floods indicate this critical period is longer. Because 3 years passed between 2017 and 2020, when our field survey was conducted, ants could have meanwhile partially recolonized the area and hence compensated for the effects of the flood. Thus, our data actually indicate that the *M. scabrinodis* ants can survive floods for less than 10 days.

A multivariate logistic model of the probability of ant presence on trap sites predicts that this probability increases with time that passed from flood. The odds ratio for ant presence increases by around 30% each year (32% in the first and 27% in the second model). The probability of ant presence also declines with longer flood duration (odds ratio for 1 day = 0.85) or distance from the nearest refuge (odds ratio for 1 m = 0.992). Comparison (divided coefficients) of function coefficients in the models indicate that in a year, ants compensate for 1.8 days of flooding with recolonization (=0.281/0.154), whereby ants colonize vacant habitats with a speed of 28.7 m per year (=0.241/0.0084; approx. 30 m).

Of all the possible GLMM (taking into account AICc), the presence of ants on a trap site is best explained by the model with two independent variables: days flooded in 2017 and distance from the nearest refuge in the 2013 flood. The model correctly classifies (meadows are/are not colonized with ants) 79.4% of all sites. Because several of the analyzed independent variables are tightly correlated, there are several other similarly good models, but all of them include only the variables days flooded and/or distance to the refuge; none of them include the variable habitat type or any interaction. The best model is presented in Table 3 and all good models (with ΔAICc < 2) are listed in the Appendix A.

## 4. Discussion

Floods and other weather-related extreme events are likely to increase with current climate change [6,33], therefore they should be accounted for in the planning and management of protected areas, particularly those where long-lasting floods are likely to become more frequent. Such floods can result in an increased mortality of terrestrial organisms, predominantly those with low mobility such as insect larval stages and some other invertebrates [7,34,35,36]. Our study was motivated by the observation of a more than tenfold decline (492 in 2008 to 27 in 2012) in the abundance of *P. teleius* adults in local metapopulation at Ljubljansko barje Plain between 2008 and 2012 [9,10], possibly affected by long-lasting flood in 2010 (Figure 1) covering most of the inhabited patches. It is, however, noteworthy that no negative effect of long-term flooding was evident in the only study directly addressing the effects of flooding on survival of the *P. teleius* conducted in Poland [37]. The ant nests in areas flooded for three weeks had a higher probability of larvae occurrence than in those unaffected by floods. However, none of the study patches was fully submerged at any time with high clumps of *Carex* sp. or *Molinia* sp. serving as potential refuges for ants. Negative effects of raised water levels were, on the other hand, recorded in related Alcon blue (*Phengaris alcon*) in the Netherlands where active prevention of drainage had a negative effect on patch occupancy [38]. In our study, we demonstrated that entire areas flooded for a longer period (more than cca 10 days) lost all the *M. scabrinodis* colonies. These floods resulted in observed mortality of *P. teleius* larvae likely either by direct effect of flooding or due to starvation as larvae cannot survive without their host ant.

In order to exclude the change in land use as a potential decline of *P. teleius*, we compared GERK layers of land use for the last 15 years in the study area [27] and no significant differences were found. Also, a slight increase in habitat suitability approximated by the presence and abundance of the host plant *Sanguisorba officinalis* was recorded during the aforementioned surveys [10], therefore the direct larval mortality or mortality of the larval host ants from the genus *Myrmica* is considered most likely reasons for the decline in adult butterfly abundance and distribution in the study area. It is known that long-lasting floods can cause *Myrmica* ant mortality [21,22]. Our results clearly corroborate these finds, as the last three long-lasting foods in the study area left some suitable habitats still devoid of the host ants. The probability of occurrence of host ants is negatively affected by the duration of floods and distance from the closest area without long-lasting floods acting as potential refugia (Figure 3). The best GLMM model correctly classified 79.4% of the studied meadows (trap sites) based on the above-mentioned criteria further exemplifying the negative effect of long-lasting floods on host ants.

When interpreting the results, it is important to bear in mind that the same ant sampling plots (may) have been flooded several times (some up to three times) and the effects of these floods are likely cumulative. The analyses used have attempted to neutralize this experimental flaw (used models assumed linear additive effects of individual floods). However, the results should still be interpreted with caution. Additionally, the refuge areas were designated solely topographically not taking into account potential local refugia such as elevated clumps of grasses, bushes, or trees not detected on lidar. Therefore, our estimates of the recolonization speed of *M. scabrinodis* ants of approximately 30 m per year should be considered the maximum recolonization speed as shorter recolonizations from local refugia could not be accounted for.

*Myrmica* ants spread mainly through budding from the existing colonies [24] and relay less on nuptial flights, therefore their colonization speeds are not directly comparable with other species of ants. In the case of an invasive South American fire ant (*Wasmannia auropunctata*) in Gabon, the estimated colonization speed was approximately 70 m per year in undisturbed forest habitat but up to 4 km in areas of human disturbance and facilitated by human activity [39]. Our approach could therefore be used in similar studies with multiple flooding for ants which spread mainly through budding from the existing colonies, but not for those spreading with nuptial flights. For a more accurate estimate of the recolonization speed of *M. scabrinodis,* a multi-year survey in partially uninhabited or uninhabited plots at the study site would be required.

Although floods in general have contrasting effects on local biodiversity, the increased mortality of less mobile organisms and potential facilitation of the spread of invasive plants [40] are outweighed by positive effects, such as increased soil humidity enabling the persistence of wet meadow communities [41], slowing down succession on abandoned meadows [24], and, most importantly, hindering of intensification of farming. Thus, we are hoping that our results will not be misrepresented as advocacy for extensive drainage, but rather to underline the need for conservation and management on larger scales, covering whole metapopulations of threatened wet meadow species specialists with the inclusion of areas less affected by long-lasting floods.

## Figures and Tables

**Figure 1 insects-14-00891-f001:**
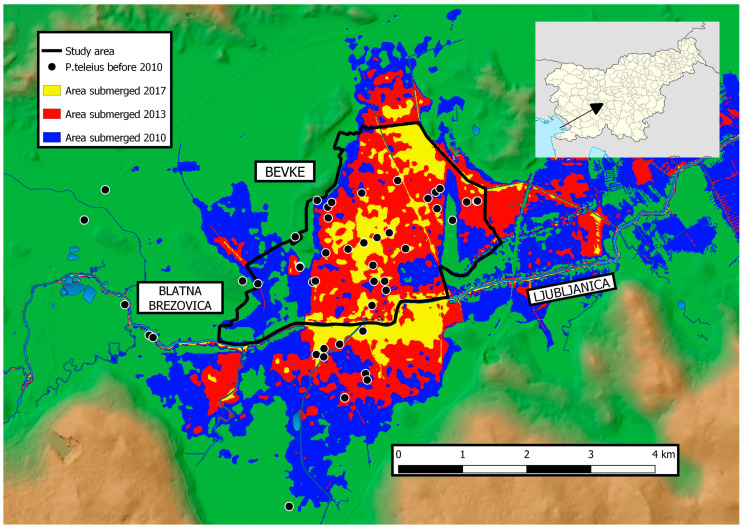
The extent of long-lasting floods since 2010 in the study area in the central part of Ljubljansko barje Plain in Slovenia. The distribution of *Phengaris teleius* before the 2010 extensive floods is extracted from [9].

**Figure 2 insects-14-00891-f002:**
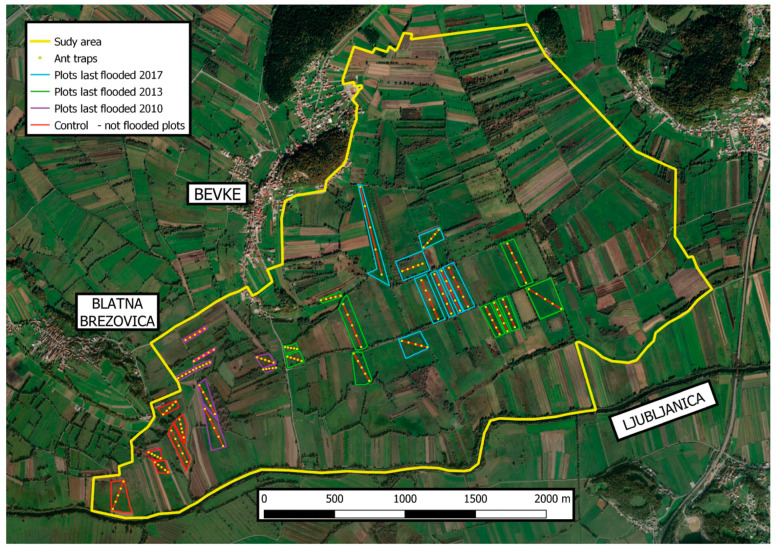
Position of meadow plots and ant traps in the studied area of Ljubljansko barje in central Slovenia. The color of plot outlines indicates the year the meadow was last flooded for 10 or more days.

**Figure 3 insects-14-00891-f003:**
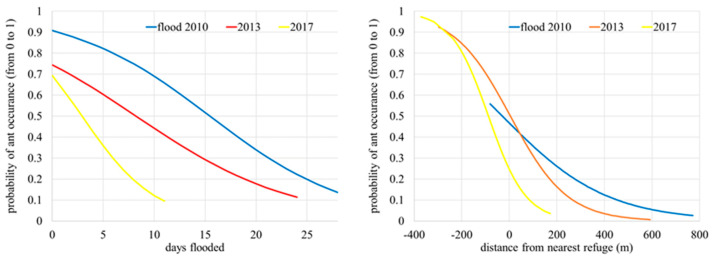
Logistic models of the presence of ants on examined trap sites with regard to the number of days the sites were flooded (**left**) and the distance of the trap site to the nearest refuge (**right**) for ants in floods in 2010, 2013, and 2017.

**Table 1 insects-14-00891-t001:** Basic statistics of all three floods in the central part of Ljubljansko barje Plain in Slovenia, estimates based on exact locations of ant traps in the survey (n = 160).

Year		Minimum	Mean	Median	Maximum
2010	days flooded	0	19	23	29
	>10 days flooded		0.79		
	distance to nearest *Myrmica* refuge *	−81	219	107	771
2013	days flooded	0	13	17	24
	>10 days flooded		0.54		
	distance to nearest *Myrmica* refuge *	−297	101	43	591
2017	days flooded	0	5	6	11
	>10 days flooded		0.34		1
	distance to nearest *Myrmica* refuge *	−372	−49	−44	172

* Distance (in meters) designates the distance of the trap site from the closest refuge in which ants are believed to have been able to survive the flood (i.e., a flood that lasted under 10 days). Positive distances are actual distances from the refuge edge. Negative distances are assigned to areas inside refuges (where *M. scabrinodis* are believed to have survived) and a higher absolute value denotes a higher distance from the refuge edge.

**Table 2 insects-14-00891-t002:** Logistic regression model of the impact of time passed (years since the flood), duration of the flood (top part of the table), and distance to the refuge (bottom part of the table) on the probability of *M. scabrinodis* presence on examined trap sites (n = 160) in floods in 2010–2017.

	Const.B0	Days Flooded	Years from Flooding
Estimate	−0.810	−0.154	0.281
Odds ratio (for unit change)	0.445	0.857	1.324
Odds ratio (range)		0.012	7.150
	**Const.B0**	**Distance from Refuge (m)**	**Years from Flooding**
Estimate	−1.844	−0.008	0.241
Odds ratio (for unit change)	0.158	0.992	1.272
Odds ratio (range)		0.000	5.394

**Table 3 insects-14-00891-t003:** Parameters of the best GLMM, which explains the probability of ant presence on examined sites determined with the best subset algorithm and AIC criterion. The model predicts that the probability of ant presence declines with the duration (number of days) of flooding of sites in 2017 and their distance to the nearest ant refuge in 2013. The model correctly classifies 79.4% of sites.

	Const.B0	Distance from Refuge in 2013	Days Flooded 2017
Estimate	0122	−0.007	−0.049
Odds ratio (unit change)	1.129	0.993	0.952
Odds ratio (range)		0.002	0.583

## Data Availability

All the data used in this article can be found by contacting the authors.

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
