# Peer review of "Impact of Long-Term Floods on Spatial Dynamics of Myrmica scabrinodis, a Host Ant of a Highly Threatened Scarce Large Blue (Phengaris teleius)"

_insects, 2023, doi:10.3390/insects14110891_

Round 1

Reviewer 1 Report

Comments and Suggestions for Authors

Author Response

Dear reviewer.

We wanted to take a moment to express our gratitude for the time and effort you put into reviewing and correcting our manuscript. Your feedback was invaluable in helping us improve our work. Thank you.

Line 47: We changed the text accordingly.

Line 111: We added the suggested reference.

Lines 187-194: We added a table as an appendex 2

Figure 2: Typo was corrected.

Reviewer 2 Report

Comments and Suggestions for Authors

Dear Authors,

I find the work interesting in its attempt to analyze a phenomenon that impacts biodiversity in general and unfortunately will be increasingly frequent and unpredictable in the near future due to ongoing climate change. In addition, although focused on the impact of a very specific host-parasite system, it takes into consideration one of the most threatened butterfly species at European level, for which these types of studies are of enormous importance in order to develop strategies for long-term management their habitats and mitigation of impacting factors.

The work is well written although I detected a series of errors which I report as minor suggestions. Although the methods applied seem appropriate to me, I believe it is necessary to clarify some aspects of the statistical analyses, including details in the text and a table with the results of the model selection. In the discussion you could also formulate hypotheses on what the direct impact of these flooding events could be on the parasite population.

You can find my comments below.

Kind regards.

Major comments

Figure 2.

-       in the legend change Plots lest flooded with last flooded.

-       It seems that in one purple meadow the traps were more than 5. Is it a mistake or was there any specific reason to place more than 5 traps?

-       Line 184-185, I think it would be essential to visualize on the map the refeges, to better understand the systems. This would also help the comparison with other phengaris sites which experience the same flooding issue.  

Line 163. Some of the meadows do not seem separate. Please, report the minimum distance that separates the closests meadows

Line 192, it is not clear how the these values were calculated. If a pitfall felt inside a refugee area, did you calculate its distance from the nearest border of the refugee area and convert it to a negative value? Please, specify better in the method section.

Also, plotting the refuges on the map would help to better understand the spatial connection between heavily flooded areas, those less impacted by floods and the refugee areas.

Statistical analyses

I think it is important to also include the software used for the analysis. In case of R, I also suggest you to include the used packages since no script code was attached to the manuscript. This would allow analyses to be replicable and reader would better follow the analyses.

Line 197, I think it would be useful to already report the correlation that were tested. Please, also specificy if the data from the 3 floods and control were treated separately

Line 234, what do you mean by habitat type? Do you mean flooded area and areas with a lower impact of floods? Please, give additional information about the habitat type.

Line 253, I suggest to use M. scabrinodis all over the manuscript since it is the only species present in the area

Line 260, why do you use variable(s)? This creates confusion. Please, be more specific.

Line 264 and… Please, include references to figure 3 and table 2 in the result section

Line 296, Please, report the list of evaluated models and their scores in supplementary material. I think it is important for the reader to visualise the tested models.

Discussion. I suggest you include in the discussion a paragraph explaining the impact of these floods on the host ant and on the social parasite Phengaris teleius, whose impact you have observed is notable.

Perhaps one might think that prolonged floods could affect the colonies in the most submerged areas, causing mortality on the individuals and consequently on the parasite.

Unlike the cuckoo species which are more integrated into the colony, during a hypothetical translocation of the flooded colony, the certainly less integrated P. teleius could be left behind.

Secondly, in a recent work (Witek et al. 2023) on M. scabrinodis and P. teleius the authors evaluated the spatial organization of colonies and parasites and show that the size of the nest is a relevant factor to explain the presence of the parasite. Consequently, it is possible that the budded colonies that manage to reach the refuge areas are not large enough to support the parasites in the following season or can support a small number of them. This effect could therefore translate into a generalized decrease in the P. teleius population over time.

I suggest you to take these aspects into consideration in the discussion.

Comments on the Quality of English Language

Dear Authors,

I report below some minor suggestions and mistake along the text.

Kind regards.

Line 5, coma is missing between authors’ names.

Line 13, unique -> specific

Line 14 of -> of the

Line 22, recolonization At – full stop is missing

Line 24, change to -> populations of the endangered

Line 51, with one the highest -> with one of the highest

Line 53, [2] The – full stop missing

Line 71, please indicate the coutry; readers may not be familiar with this area.

Line 72, aiming on prevention of floods -> aiming at prevent floods

Line 90, [15], After the -> full stop instead of coma

Line 91, reference (Sala et al. 2014) should be converted to a number

Line 92, [16], The caterpillars -> full stop instead of coma

Line 98, Barje was previously reported with uncapitalised initial – Please, be consistent through the text

Line 99, remove Verovnik

Line 100,137 , the first part of the sentence is in a different font character or size. This happens in other part of text. Please, verify along the text.

Line 117, unsing above -> using the above

Line 118, remove “of ants”

Line 125, 175, Myrmica in italics

Line 141, the phengaris -> phengaris

Line 143, therefore -> Therefore

Line 146, aforamentioned -> the aforamentioned

Line 183, flooded less -> flooded for less

Line 226, any a better -> any better

Line 258, probably a full stop is missing between meadow and 12

Line 270, floods so these -> floods; so these

Line 271, floods -> flood events

Line 333, three long lasting foods in the -> three long-lasting floods in the

Author Response

Dear reviewer.

We wanted to take a moment to express our gratitude for the time and effort you put into reviewing and correcting our manuscript. Your feedback was invaluable in helping us improve our work. Thank you.

Major comments

Com 1: Fig 2 - in the legend change Plots lest flooded with last flooded.

We changed the text accordingly.

Com 2: It seems that in one purple meadow the traps were more than 5. Is it a mistake or was there any specific reason to place more than 5 traps?

In this case two meadows are quite close to each other and might be confused as one in the map, therefore it looks like there are more traps inside. All meadows have equal sampling effort (5).

Com 3: Line 184-185, I think it would be essential to visualize on the map the refeges, to better understand the systems. This would also help the comparison with other phengaris sites which experience the same flooding issue.

The boundaries of the refuge mostly follow the boundaries of the flooded areas in each of the three floods and are therefore 3 different in entire study period. We are afraid that if we were to add refuge lines, the picture, which is already very full of information, would become too opaque. We therefore suggest it is better to keep the figure as it is, but we have made it clearer in the text what the refuge boundaries are, so we hope depicting them is now less needed.

Com 4: Line 163. Some of the meadows do not seem separate. Please, report the minimum distance that separates the closests meadows

We added information on the minimum distance among meadows/plots.

Com 5: Line 192, it is not clear how the these values were calculated. If a pitfall felt inside a refugee area, did you calculate its distance from the nearest border of the refugee area and convert it to a negative value? Please, specify better in the method section.

We extended our explanation to make it more clear: Distance from the edge of nearest refuge to each ant trap (in meters, for each flood event separately); for the traps set outside refuges the distance is a positive value and for the traps within refuges the values are negative.

Com 6: Also, plotting the refuges on the map would help to better understand the spatial connection between heavily flooded areas, those less impacted by floods and the refugee areas.

Please see our answer on the comment 3.

Com 7: Lines 187-194: Although the description of variables was clear, a table would be helpful

We added the table as an appendix.

Statistical analyses

Com 8: I think it is important to also include the software used for the analysis. In case of R, I also suggest you to include the used packages since no script code was attached to the manuscript. This would allow analyses to be replicable and reader would better follow the analyses.

Added as suggested

Com. 9: Line 197, I think it would be useful to already report the correlation that were tested. Please, also specificy if the data from the 3 floods and control were treated separately

The information on used variables and thus tested correlations is provided in previous chapter, so we would prefer not to repeat it again here as this would extend the MS. Meaningful results are reported in Results chapter. We are not sure we understand the second part of this comment. Data/variable handling is explained in the Methods chapter that follows this line.

Com 10: Line 234, what do you mean by habitat type? Do you mean flooded area and areas with a lower impact of floods? Please, give additional information about the habitat type.

Thank you, this was our error, we meant habitat use (explained in previous chapter) and have now corrected in MS.

Com 11: Line 253, I suggest to use M. scabrinodis all over the manuscript since it is the only species present in the area

We agree and have applied the recommended change throughout the text. 

Com 12: Line 260, why do you use variable(s)? This creates confusion. Please, be more specific.

We agree and have corrected. There is just one dependent variable. Use of plural slipped our attention and ended up here from the initial exploration of possible dependent variables (binary, continuous representation of ant presence).

Com 13: Line 264 and… Please, include references to figure 3 and table 2 in the result section

The references to figure 3 and table 2 were added.

Com 14: Line 296, Please, report the list of evaluated models and their scores in supplementary material. I think it is important for the reader to visualise the tested models.

As suggested list of models and their scores are added as supplementary material (Appendix 1).

Discussion

Com 15: I suggest you include in the discussion a paragraph explaining the impact of these floods on the host ant and on the social parasite Phengaris teleius, whose impact you have observed is notable.

We added the following explanation: In our study we showed, that entire areas which were flooded for more than 10 days lost all the M. scabrinodis colonies. This has likely led to observed mortality of P. teleius larvae either by direct effect of flooding or due to starvation as larvae cannot survive without their host ant.

Com 16 Perhaps one might think that prolonged floods could affect the colonies in the most submerged areas, causing mortality on the individuals and consequently on the parasite.

See the above added explanation.

Com 17, Unlike the cuckoo species which are more integrated into the colony, during a hypothetical translocation of the flooded colony, the certainly less integrated P. teleius could be left behind.

As stated in the above addition we presumed all ant colonies were destroyed by flooding as translocation/evacuation is not known to be a mechanism of flood avoidance in any Myrmica sp. We are not sure we understand this suggestion/comment.

Com 18 Secondly, in a recent work (Witek et al. 2023) on M. scabrinodis and P. teleius the authors evaluated the spatial organization of colonies and parasites and show that the size of the nest is a relevant factor to explain the presence of the parasite. Consequently, it is possible that the budded colonies that manage to reach the refuge areas are not large enough to support the parasites in the following season or can support a small number of them. This effect could therefore translate into a generalized decrease in the P. teleius population over time.

Yes, we are aware of this study, but as stated above the Myrmica ants do not flee from flooded areas or bud out of flooded areas (maybe those on the very edge of the flood), and the colonies are locally destroyed. The refuges in our case are considered those areas where host ants are present and were not affected by flooding (and host a source population for spatial recolonization of flooded areas). These unflooded upland areas thus represent the origin of recolonization of the flooded areas and thus also for gradual return of the butterfly.

Comments on the Quality of English Language

Dear Authors,

I report below some minor suggestions and mistake along the text.

Kind regards.

Line 5, coma is missing between authors’ names.

Corrected.

Line 13, unique -> specific

Corrected.

Line 14 of -> of the

Corrected.

Line 22, recolonization At – full stop is missing

Corrected.

Line 24, change to -> populations of the endangered

Corrected.

Line 51, with one the highest -> with one of the highest

Corrected.

Line 53, [2] The – full stop missing

Corrected.

Line 71, please indicate the coutry; readers may not be familiar with this area.

Corrected.

Line 72, aiming on prevention of floods -> aiming at prevent floods

Corrected.

Line 90, [15], After the -> full stop instead of coma

Corrected.

Line 91, reference (Sala et al. 2014) should be converted to a number

Corrected.

Line 92, [16], The caterpillars -> full stop instead of coma

Corrected.

Line 98, Barje was previously reported with uncapitalised initial – Please, be consistent through the text

Corrected.

Line 99, remove Verovnik

Corrected.

Line 100,137 , the first part of the sentence is in a different font character or size. This happens in other part of text. Please, verify along the text.

Corrected.

Line 117, unsing above -> using the above

Corrected.

Line 118, remove “of ants”

Corrected.

Line 125, 175, Myrmica in italics

Corrected.

Line 141, the phengaris -> phengaris

Corrected.

Line 143, therefore -> Therefore

Corrected.

Line 146, aforamentioned -> the aforementioned

Corrected.

Line 183, flooded less -> flooded for less

Corrected.

Line 226, any a better -> any better

Corrected.

Line 258, probably a full stop is missing between meadow and 12

Corrected.

Line 270, floods so these -> floods; so these

Corrected.

Line 271, floods -> flood events

Corrected.

Line 333, three long lasting foods in the -> three long-lasting floods in the

Corrected.

Round 2

Reviewer 2 Report

Comments and Suggestions for Authors

Dear Authors,

I am satisfied with the revisions made. I found some small typos throughout the text which I report below.

Kind regards.

line 15, Present study -> The present study

Line 164, 12.5m -> 12.5 m

Line 195, Myrmica in italics

Line 267 [32], -> [32].

Line 282, All ants were identified at the species level. I would move this sentence to the method part. 

Line 328, Should it be AICc?

Line 335, Best model -> The best model

In the legend of Appendix 2, use Only instead of Just. It is slightly more formal.

Author Response

We agree with all suggestions and have incorporated them throughout the manuscript. Thank you again for your efforts.
